**Cite this article:** Sadras VO. 2021
Evolutionary and ecological perspectives on the
wheat phenotype. *Proc. R. Soc. B* **288**:
20211259.

plant science, ecology, biological applications

technology, theory, yield, aphids, osmotic
stress, water-soluble carbohydrates

**Author for correspondence:**
Victor O. Sadras
e-mail: victor.sadras@sa.gov.au

# Evolutionary and ecological perspectives on the wheat phenotype

Victor O. Sadras

South Australian Research and Development Institute, and School of Agriculture, Food and Wine,
The University of Adelaide, Australia

  VOS, 0000-0002-5874-6775

Technologies, from molecular genetics to precision agriculture, are outpacing theory, which is becoming a bottleneck for crop improvement. Here, we outline theoretical insights on the wheat phenotype from the perspective of three evolutionary and ecologically important relations—mother–offspring, plant–insect and plant–plant. The correlation between yield and grain number has been misinterpreted as cause-and-effect; an evolutionary perspective shows a striking similarity between crop and fishes. Both respond to environmental variation through offspring number; seed and egg size are conserved. The offspring of annual plants and semelparous fishes, lacking parental care, are subject to mother–offspring conflict and stabilizing selection. Labile reserve carbohydrates do not fit the current model of wheat yield; they can stabilize grain size, but involve trade-offs with root growth and grain number, and are at best neutral for yield. Shifting the focus from the carbon balance to an ecological role, we suggest that labile carbohydrates may disrupt aphid osmoregulation, and thus contribute to wheat agronomic adaptation. The tight association between high yield and low competitive ability justifies the view of crop yield as a population attribute whereby the behaviour of the plant becomes subordinated within that of the population, with implications for genotyping, phenotyping and plant breeding.

## 1. Introduction

> …the reservoir of theory was being drained. Technological progress would begin to decelerate and eventually come to a complete halt… Cixin Liu, The Dark Forest [1, p. 275]

Technologies, from molecular genetics to precision agriculture, are outpacing theory, which is becoming a bottleneck for crop improvement and agronomy [2,3]. This is part of a broader problem in biology whereby we are 'now generating gigantic amounts of genomic, proteomic, metabolomic and physiomic data. We are swimming in data. The problem is that the theoretical structures within which to interpret it are underdeveloped or have been ignored and forgotten. There is an essential incompleteness in biological theory that calls out to be filled' [4, p. VIII].

The phenotype includes all traits of an organism other than its genome [5]; thus, grain yield, seed storage proteins, susceptibility to rust, stomatal conductance, nitrogen uptake and root architecture are all agronomically important aspects of the crop phenotype. West-Eberhard's book *Developmental plasticity and evolution* is a milestone in the contemporary theory of the phenotype [5]. Other substantive theoretical insights had a narrower focus [6–11] but all aligned with Dobzhansky's [12] premise of an evolutionary perspective, often combined with developmental and ecological angles. A developmental perspective reveals the inadequacy of the unidirectional cause-and-effect arrow from genotype to phenotype as the same genome returns more than 30 cellular phenotypes in plants and over 200 in humans [5,13,14]. In this developmental context, the notion of downward causation, where higher scales of organization can causally influence behaviour at lower scales, is useful to understand and formalize the phenotype [13,15–17].

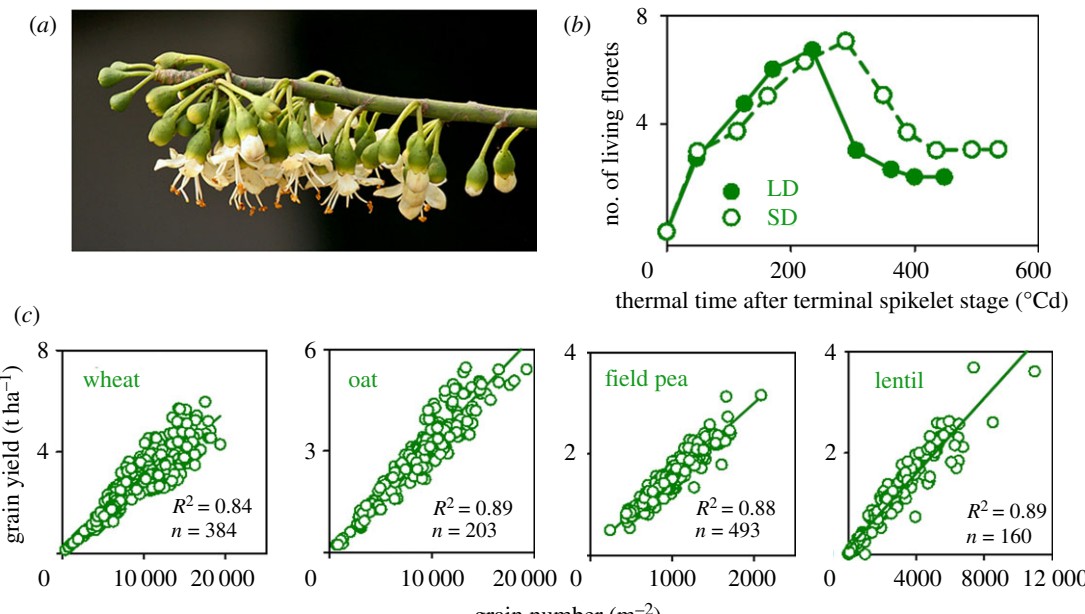

**Figure 1.** Angiosperms overproduce flowers and ovules and annual crops respond to environmental variation through grain number. (*a*) *Ceiba petandra*, native to Mexico, Central America and the Caribbean, typically produces 1000 or more flowers for every mature fruit [23]. (*b*) Dynamics of floret survival in wheat illustrate overproduction and mortality. SD is short day and LD is long day, suggesting the plant might be using photoperiod to anticipate grain fill conditions. (*c*) Annual crops respond to environmental variation through grain number, which accounts for most of the variation in yield; average grain weight accounts for the residual variation. Data sources: (*b*) [24], (*c*) wheat [25], oat [26], field pea [27] and lentil [28]. Photograph (*a*) is from Wikipedia (https://en.wikipedia.org/wiki/Ceiba_pentandra) accessed on 11 June 2021. (Online version in colour.)

Darwin [18. p. 69] noted the dominant role of abiotic factors as drivers of evolution in the most extreme environments, '…the Artic regions, or snow-capped summits, or absolute deserts'. Elsewhere, he argued, the relation of organism to organism is the most important of all relations as verified in contemporary models [19–22]. Here, we outline theoretical insights on the wheat phenotype from the perspective of three evolutionary and ecologically important relations—mother–offspring, plant–insect and plant–plant.

## 2. Annual crops and fishes respond to environmental variation through offspring number

Angiosperms overproduce flowers and ovules (figure 1*a,b*) [23]. A handful of non-mutually exclusive hypotheses explain an overproduction of flowers and ovules, including compensation for the loss of developing embryos, anticipation of favourable conditions for fruit and seed set (e.g. availability of pollinators and resources), selective abortion of low-quality embryos and uniform seed production mediated by the selection of fertilized ovules with similar resource absorption rates [23,29,30]. Next, we observe that annual crops respond to environmental variation through grain number (GN), which accounts for most of the variation in yield between crop failure and potential yield (figure 1*c*); exceptions are rare.

The strong correlation between yield and GN (figure 1*c*) had been misinterpreted as cause-and-effect until Sinclair & Jamieson [31] challenged this view. They used a metaphor: the number of bottles used by the brewery in marketing its beer correlates with, but is an unlikely cause of, total beer production; they focused on resources driving both GN and yield. Their proposition had gaps and triggered controversy

[32,33], but motivated further interpretations of the correlation between yield and GN. Why would a plant adjust GN and keep grain size (GS) stable?

An excursion into evolutionary territory showed a striking similarity between crop and fishes (figure 2*a*). The question changed fundamentally—we are now asking what wheat and salmon have in common [38,39]. In contrast with most mammals, birds and social Hymenoptera where parents provision their offspring as they develop, wheat and salmon offspring are on their own after birth, and the size of the seed and egg are subject to mother–offspring conflict and stabilizing selection as outlined in the model of Smith & Fretwell [37] (figure 2*b*). Building on this model with explicit consideration of genomic conflict, De Jong *et al.* [40] predicted that (i) when offspring genes drive the provisioning of the seed, the optimal seed size can be calculated with Hamilton's rule, and (ii) when seed size is a compromise between mother and offspring, selfers such as wheat would produce smaller seed than outcrossing plant species. The original model [37] did not consider factors such as environmental variability [41,42], variance in fertilization success among flowers within a plant [43], density-dependent mechanisms [44], maintenance respiration [45] and overhead cost of reproduction [46]. Nonetheless, the core principle remains: mother–offspring conflict emerges because, beyond a certain size, the maternal fitness benefit from larger offspring is offset by the benefit from creating and provisioning additional offspring (figure 2*b*). Answers to the wheat–salmon question progressed in theoretical studies of grain yield in annuals [39,47–49] accounting for genomic conflict [50], the evolution in the units of selection [51] and hierarchies of plasticity [52]. The role of labile reserve carbohydrates, routinely quantified as water-soluble carbohydrates (WSC) in cereal shoot [53–56], remains a major gap.

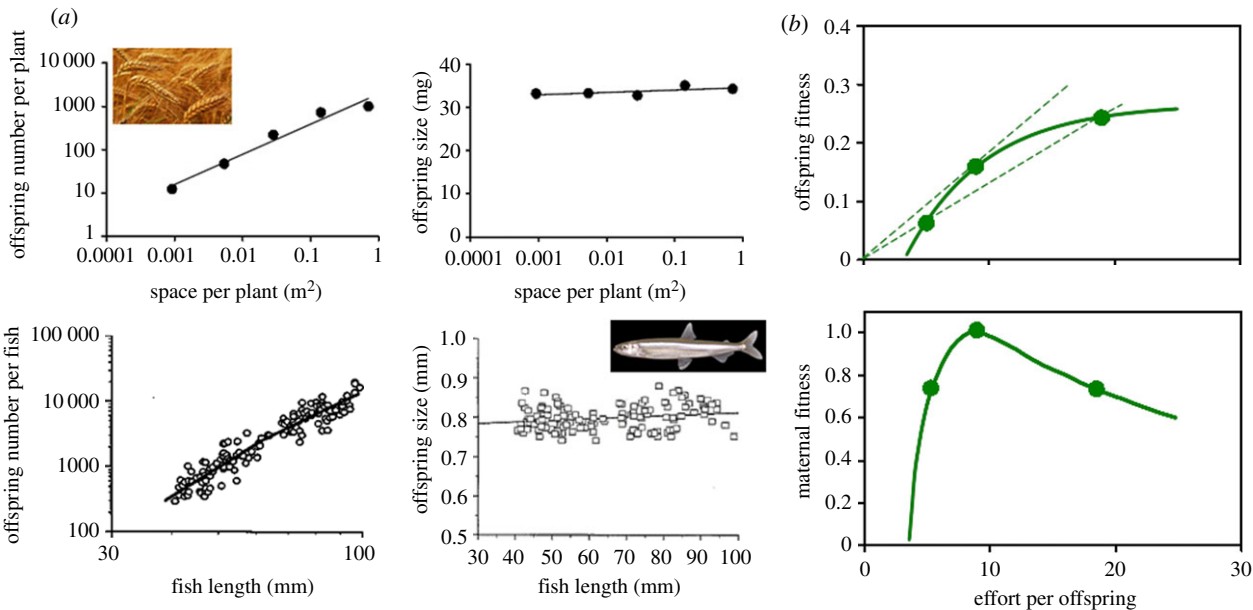

**Figure 2.** In species that lack parental care, offspring size is subject to mother–offspring conflict and stabilizing selection. (*a*) Wheat (*Triticum aestivum,* top) and pond smelt (*Hypomesus nipponensis,* bottom) adjust offspring number and conserve offspring size in response to availability of resources. Note the log scale necessary to capture the variation in offspring number. Wheat and pond smelt are both semelparous, that is, they reproduce once and die. For iteroparous organisms, like perennial plants and most mammals, lifetime reproduction is divided into many discrete bouts, hence evolutionary explanations for the trade-off between offspring size and number require more complex models accounting for the effect of reproductive effort on any one period on further survival and reproduction [34]. (*b*) Relationship between offspring fitness and maternal fitness (dimensionless) and allocation of resources to offspring. The model considers a range of maternal strategies investing a fixed amount of resources (1000 units) among a variable number of offspring, hence the variable effort per offspring (*x*-axis), which for our purposes could be approximated to seed or egg size. The dashed lines in the top diagram represent two adaptive functions intersecting the curve of possible maternal types; the intersection involving the adaptive function of highest slope corresponds with the optimal maternal type defined in terms of effort per offspring. Sources: (*a*) wheat [35], and pond smelt [36] and (*b*) [37]. (Online version in colour.)

## 3. The puzzling role of labile reserve carbohydrates in the algorithm of wheat reproductive allocation and grain yield

Yield is the product of GN per unit land area and average grain weight, but these traits are not independent [48,57]. The simplest algorithm accounting for the simultaneous determination of GN and potential GS is

$$\frac{R}{GS} \approx GN, \tag{3.1}$$

where $R$ is crop resources. Theory and empirical observations justify using crop growth rate (CGR) in the critical period of grain set as a surrogate for $R$ in annual crops [57,58] including wheat [59]. The rationale of this model involves four main elements [39,47–49,57]. The first assumption is that plants account, albeit imperfectly, for past, current and future environmental conditions by combining transgenerational mechanisms including epigenetics, proximate environmental cues such as direct sensing of water and nutrient availability in soil, and cues such as photoperiod that allow for future conditions [49] (e.g. figure 1*c*). Second, ovary size sets the upper limit of GS, and this process is simultaneous with the critical period of grain set [59]. Third, conserved offspring size is adaptive; it has typically high heritability, e.g. median of 52 reports = 0.78 [39]. Fourth, the plant responds to environmental variation by allowing for the allocation of a variable amount of resources $R$ to GN grains of target size GS.

This model overlooks allocation to root [55] and storage of labile carbohydrates in shoot [56] that are concurrent with the determination of GN and potential GS [59]. Hence, using

CGR between stem elongation and anthesis (g m$^{-2}$ d$^{-1}$) as a surrogate for $R$ [57–59], we can rewrite equation (3.1):

$$\frac{CGR}{GS} \times E_{GR-GN} \approx GN, \tag{3.2}$$

where $E_{GR-GN}$ is the efficiency of conversion of growth rate per unit GS into GN (d$^{-1}$). Crops with more resources allocated to reserve will have lower efficiency to produce grain, hence the expected inverse function:

$$E_{GR-GN} \approx f\,(WSC)^{-1}, \tag{3.3}$$

where WSC is the amount of WSC stored in the shoot at anthesis (g m$^{-2}$). Likewise, $E_{GR-GN}$ is expected to decline with an increasing allocation of resources to root. Fruiting efficiency, defined as the number of grains per unit spike dry matter at anthesis, is an important source of variation in GN [59,60] and could be expected to contribute to $E_{GR-GN}$:

$$E_{GR-GN} \approx f\,(fruting\ efficiency). \tag{3.4}$$

We tested the predictions in equations (3.2) and (3.3) with data from an experiment with 13 historic wheat varieties adapted to winter-rainfall environments grown in two locations; equation (3.4) remains to be tested. Data conform to expectations: GN is proportional to growth rate per unit GS (equation (3.2), figure 3*a*), and high allocation to labile carbohydrates reduces the efficiency of grain set (equation (3.3), figure 3*b*). Independent, comprehensive studies under robust agronomic conditions support the conclusion that labile reserve carbohydrates may buffer grain weight under stress, but are neutral or negative for yield [53,54,63]. Large amounts of labile reserve carbohydrates may remain in mature cereal

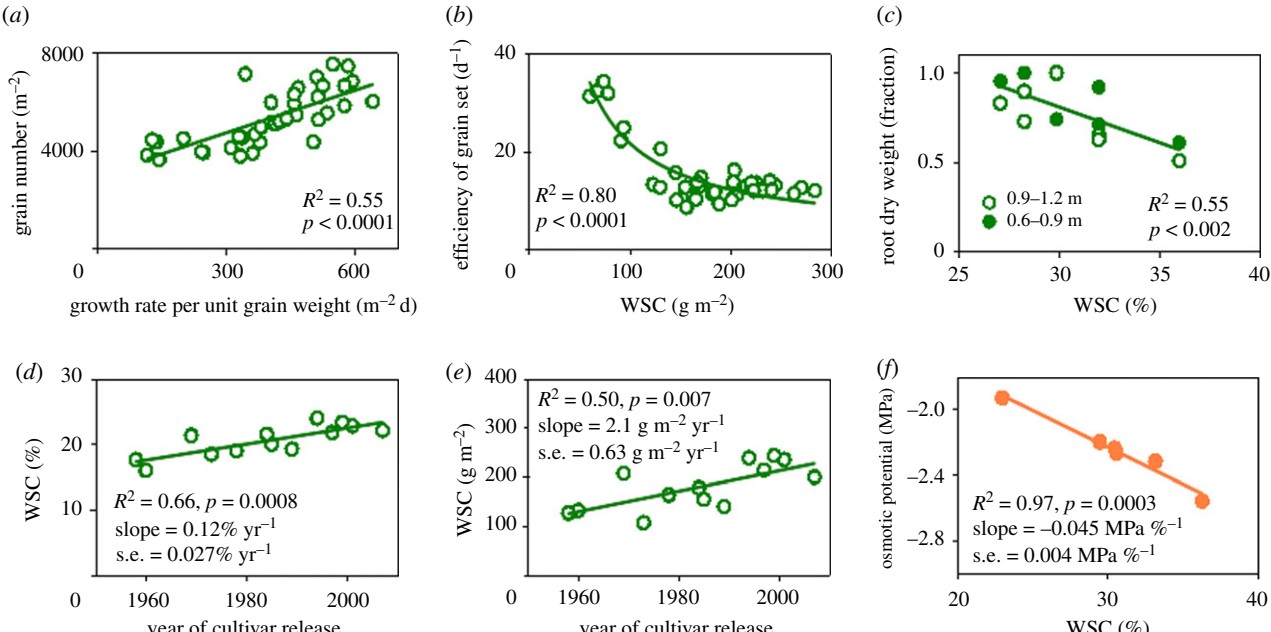

**Figure 3.** Labile reserve carbohydrates are central to the carbon economy of the crop and have overlooked implications for osmotic potential. (*a*) Wheat grain number is proportional to growth rate per unit grain weight (equation (3.2)). (*b*) The efficiency of grain set is inversely related to the amount of WSC in shoot at flowering (equation (3.3)). (*c*) Trade-off between wheat root growth and concentration of stem WSC; roots were measured in two soil layers (0.6–0.9 and 0.9–1.2 m) and biomass is relative to maximum. Selection for wheat yield and agronomic adaptation over five decades steadily increased (*d*) the concentration and (*e*) the amount of WSC in shoot at flowering. (*f*) Correlation between the osmotic potential and concentration of WSC in wheat plants. Data sources: (*a,b,d,e*) 13 Australian varieties released between 1958 and 2006 grown in two to three locations [61]; (*c*) eight wheat genotypes randomly selected from a mapping population derived from a Seri/Babax cross [55]; (*f*) three wheat cultivars infested with Russian wheat aphid (*Diuraphis noxia*) and uninfested controls [62]. (Online version in colour.)

crops, particularly under favourable growing conditions [26]. Empirical evidence also supports the trade-off between storage of labile reserve carbohydrates and root growth (figure 3*c*). On the other hand, selection for yield and agronomic adaptation over several decades has favoured a higher amount and concentration of WSC in wheat adapted to low-yielding Australian environments (figure 3*d,e*). In a historic collection of wheat cultivars adapted to high-yielding environments in the UK, the concentration of WSC in shoot at anthesis was high (greater than 41%) and did not change with the year of release, but the total amount of WSC increased at 4.6 g m$^{-2}$ yr$^{-1}$ with selection for yield between 1972 and 1995 [64]. How, therefore, has selection for yield and agronomic adaptation favoured traits—amount and concentration of labile reserve carbohydrates—that are at best neutral for yield. Correlative variation of traits is central to evolution [18,65–67]; we expect the strong directional shift in wheat labile carbohydrates (figure 3*d,e*) correlates with yet unrecognized, agronomically important traits.

Physiological, ecological and agronomic studies mostly focus on the role of reserve carbohydrates in the carbon economy of the plant, for example, as buffers for reproduction or regrowth after herbivory or fire [53–56,68,69]. Within physiological limits, labile carbohydrates have a significant osmotic effect (figure 3*f*) that has received less attention in the context of ecological and agronomic adaptation.

## 4. Selection for tolerance to aphids might favour high concentration of labile reserve carbohydrates

…plants and animals, most remote in the scale of nature, are bound together by a web of complex relations… (Darwin [18], p. 73)

Aphids (Hemiptera, Aphidoidea), and the viruses they carry, are major pests of wheat and a focus of breeding programmes worldwide [70,71]. We speculated that selection of wheat phenotypes with lower aphid load or less severity of the viral disease may have favoured the steady increase in WSC (figure 3*d,e*) that potentially challenges osmoregulation in aphids [63]. Aphids have evolved intricate anatomical, physiological and behavioural traits for osmoregulation [72–77]. An early study with *Myzus persicaea* grown on sea aster (*Aster tripolium*) found that the osmotic pressure of the excreted honeydew was similar to that of the haemolymph, thus demonstrating the aphid's ability to reduce the osmotic pressure of the ingested sap [72]. Aphid growth and fitness feature a sweet spot in response to diet's sugar concentration [76]. This is illustrated in figure 4*a*, showing the relative growth rate of pea aphid (*Acyrthosiphon pisum*) was impaired by reduced feeding reflecting the role of sucrose as a phagostimulant at low dietary sucrose concentrations, and by osmoregulation failure at high concentrations. Up to a threshold of $1.06 \pm 0.21$ M sucrose in the diet, the osmotic pressure of the aphid's haemolymph was maintained, but osmoregulation broke down above this threshold (figure 4*b*). Furthermore, the abundance of symbiotic bacteria *Buchnera* spp., critical to the supply of essential amino acids to the aphid, collapsed after a threshold of $0.87 \pm 0.18$ M sucrose in the diet (figure 4*c*). Aphids occasionally consume the dilute xylem sap, a behaviour associated with both dehydration and osmotic stress in non-dehydrated insects [77]. *Macrosiphum euphorbiae*, a common pest in potatoes, increased the time actively sucking xylem sap with both increased osmotic potential of the artificial diet and deprivation of primary symbionts with antibiotics, a condition that leads to higher haemolymph osmotic potential [77].

An aphid reproduction experiment showed a decline in the number of adult bird cherry-oat aphids *Rhopalosiphum padi*

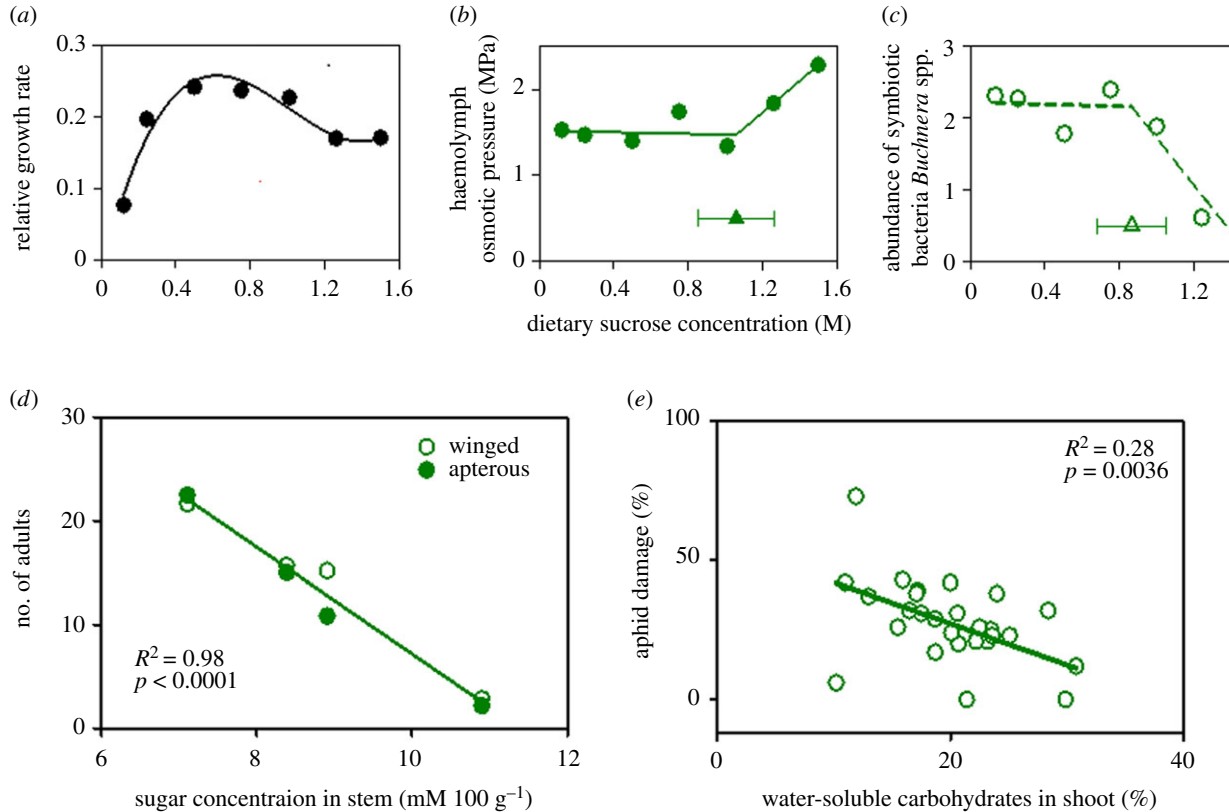

**Figure 4.** High concentration of sugars in diet disrupts aphid osmoregulation and fitness and may be adaptive for cereals. (*a*) Relative growth rate of the pea aphid *A. pisum* as a function of sucrose concentration in artificial diet. Relative growth rate is $\log_e$(day-8 mass/day-6 mass)/2], with aphids weighed on day 6 and day 8 to the nearest μg. (*b*) Osmotic pressure of the haemolymph of 8-day-old aphids reared on diets with varying concentration of sucrose. (*c*) Abundance of symbiotic bacteria *Buchnera* spp. in 8-day-old aphids on diets of varying concentration of sucrose. Abundance is $10^{-6}\times$ the number of copies of *Buchnera* dnak gene per ng total DNA. In (*b*,*c*), triangles are the inflection points (±s.e.) of the fitted curves. (*d*) The number of adult bird cherry-oat aphid (*Rhopalosiphum padi*) adults decreased with increasing concentration of sugars in wheat stem. (*e*) Russian wheat aphid (*D. noxia*) caused less damage in field-grown cereals with higher concentration of WSC in shoot. Sources of data: (*a–c*) [76], (*d*) [78], (*e*) [79]. (Online version in colour.)

with increasing concentration of sugars in wheat stem (figure 4*d*). In a field comparison of 28 cereal genotypes (15 bread wheat, five durum wheat, eight barley), plant damage caused by *Diuraphis noxia* declined with increasing concentration of WSC in shoot (figure 4*e*). Empirical evidence linking labile reserve carbohydrates and aphid tolerance is just emerging, but the view of an ecological role of labile carbohydrates opens a new dimension to the established, narrow focus on plant carbon balance and source–sink relations.

## 5. Crop yield is a population attribute whereby the behaviour of the plant becomes subordinated within that of the population

> …the struggle almost invariably will be most severe between the individuals of the same species, for they frequent the same districts, require the same food, and are exposed to the same dangers … (Darwin [18, p. 75]).

Since the inception of agriculture in the Neolithic until the end of the eighteenth century, crop yield has been measured as the ratio of seed harvested to seed sown [80], e.g. small grain crops in Europe yielded four to seven seeds per seed in the 1770s [81]. This measure of yield favoured competitive, tall plants with large root system and profuse branching. Only recently on a historical time scale, the definition of yield shifted to the current measure of mass of seed per

unit land area [80]. The selective pressure thus shifted to favour a 'communal' phenotype [82–84]. An updated evolutionary focus of crop yield in relation to plant–plant relations emphasizes kin selection and multi-level selection [85–89]. Empirical evidence supports the association between high yield per unit area and less competitive phenotypes in morphologically and physiologically diverse annual and perennial crops [90–92], including wheat (figure 5). In a collection of elite wheat CIMMYT[1] cultivars, removing or bending adjacent plant rows to relax competition with the focal central row showed a strong negative correlation between grain yield and response to reduced competition (figure 5*a*). Next, an association mapping panel of 287 CIMMYT elite lines was phenotyped for response to competition, based on the yield difference between outer and inner rows of experimental plots, to identify genomic regions associated with low competitive ability and high yield per unit area [95]. Selection for yield in low-rainfall environments of Australia over five decades returned high-yielding phenotypes with a reduced competitive ability (figure 5*b*,*c*). Binary mixtures of wheat cultivars demonstrated a strong symmetry in yield response to neighbour (figure 5*d*); this is, Halberd (the oldest, more competitive cultivar in the series) increased yield by approximately 17% with Scepter (the newest, less competitive) neighbour in comparison to pure stands, and Scepter reduced yield in a similar proportion when grown with a Halberd neighbour (figure 5*d*). All 12-pairwise combinations of cultivars grown under eight environmental

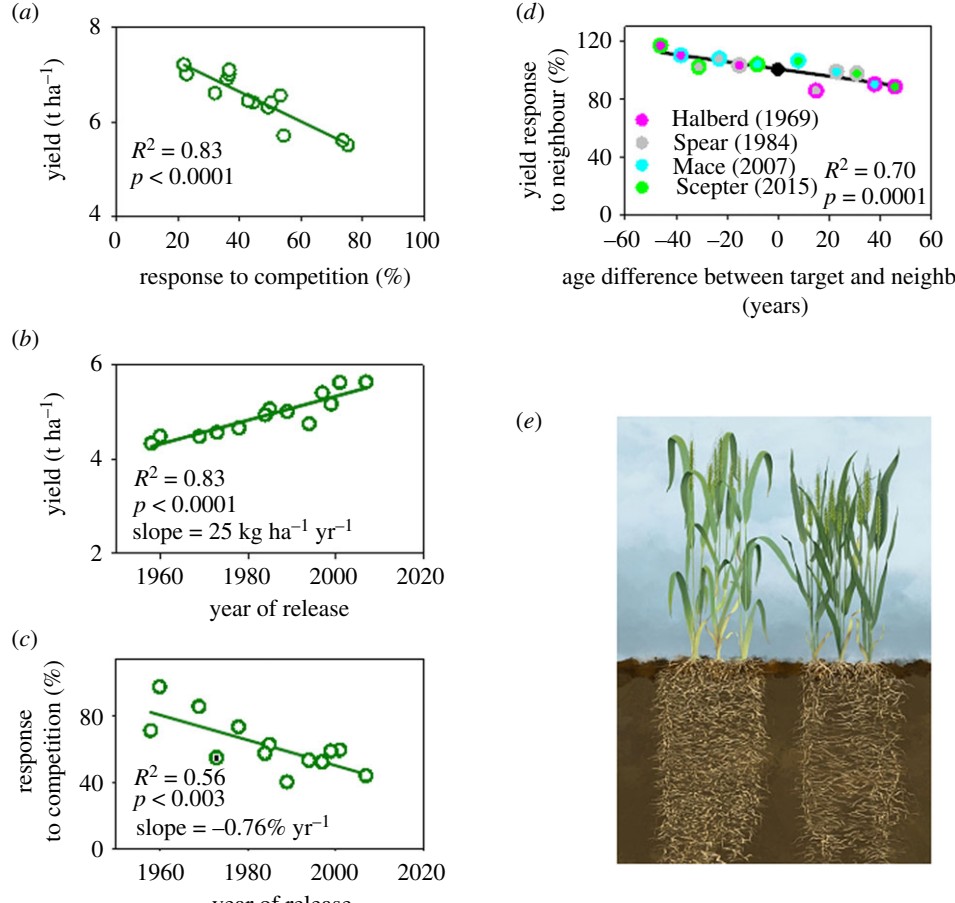

**Figure 5.** Crop yield is a population attribute whereby the behaviour of the plant becomes subordinated within that of the population. (*a*) High yield associates with a less competitive wheat phenotype in a collection of elite CIMMYT lines. (*b*) Five decades of breeding effectively increased the yield of wheat varieties adapted to low-rainfall environments of Australia and (*c*) selective pressure for yield favoured less competitive phenotypes. In (*a*) and (*c*), response to competition is the ratio of yield in crops with full competition in centre rows of experimental plots, and yield where competition has been relaxed by manipulation of adjacent rows. (*d*) Binary mixtures of wheat varieties released between 1969 and 2015 show symmetric yield response to competition. The central colour of the symbol shows the target variety where yield was measured and the edge colour is the neighbour variety; for example, the pink symbol with green edge is the yield of Halberd with Scepter neighbour as a percentage of the yield of Halberd in pure stand. The black symbol is yield in pure stands. (*e*) Comparison of a phenotype with high yield and low competitive ability (right) and its counterpart with low yield and high competitive ability (left). Sources of data: (*a*) [93], (*b,c*) [61], (*d,e*) [94]. (Online version in colour.)

conditions aligned in a plot of yield of target cultivar relative to pure stand versus the age difference between target and neighbour (figure 5*d*). Owing to the steady selection pressure returning a linear genetic yield gain (figure 5*b*), difference in year of release between target and neighbour, for example, 46 years between Halberd and Spear, roughly captures the genetic divergence between cultivars. Figure 5*e* updates the communal wheat phenotype [94]. The less competitive, higher yielding phenotype is shorter and intercepts less radiation. Higher radiation use efficiency compensates for the lower interception of radiation in the less competitive phenotype and relates to an erectophyl canopy that favours more radiation and higher nitrogen concentration in leaves at the bottom of the canopy. The less competitive phenotype has a smaller root system with compensatory higher nitrogen uptake per unit root length (figure 5*e*).

The theoretical and empirical evidence for the tight link between high yield and low competitive ability justifies the view of crop yield as a population attribute whereby the behaviour of the plant becomes subordinated within that of the population [96]. Two implications from this conclusion illustrate how data-driven technologies for crop improvement would

benefit from interpretations of the crop phenotype informed by evolutionary, ecological and developmental perspectives.

First, a better understanding of trade-offs, which can represent either constraints or opportunities, is key to understanding past progress, remaining opportunities and the ultimate limits to crop genetic improvement [97,98]. Plant breeding is unlikely to improve traits shaped by natural selection over evolutionary timescales, such as the efficiency of photosynthetic enzymes [87,99], but unrealized opportunities may exist for the selection of traits that increase crop yield at the expense of plant fitness [87,97,100]—plant breeding should be based on group selection [89]. Nonetheless, crop adaptation to current agricultural environments can also be achieved at the expense of adaptation to early environments. For example, the trade-off between specificity and reaction rate of rubisco is a constraint for the improvement in the efficiency of rubisco by reducing photorespiration, but $CO_2$-specificity becomes less important with increasing concentration of atmospheric $CO_2$ [100].

Second, phenotyping and genotyping efforts must account for plant–plant relations [2,3]. Traits such as herbicide tolerance usually scale from plants in controlled environments to

agronomic conditions. But yield and photosynthesis typically do not [99,101]. In a textbook study, chlorophyll-deficient soybean isolines, Clark y9 and Clark yu, featured about half the concentration of *leaf* chlorophyll in comparison to the normal pigmented wild-type Clark [102]. Despite this massive handicap, the canopies of the mutant isolines were photosynthetically similar or out-performed the wild-type; this was attributed to more radiation penetrating down to the lower leaves in the chlorophyll-deficient canopies, effectively increasing the leaf area contributing to *canopy* photosynthesis [102]. Indeed, gene expression and the phenotype depend on both stand density and genetic identity of neighbouring individuals [85,103–106]. Overlooking plant–plant relations is a source of inefficient plant phenotyping, even under controlled conditions where size-hierarchies develop from interference between neighbours [107].

## 6. Conclusion

Data-driven approaches to improve crops are powerful but incomplete. Theoretical perspectives promoted major conceptual twists bringing the trade-off between GN and grain weight into the framework of mother–offspring conflict, and expanding the role of labile reserve carbohydrates from a simple component of the plant carbon balance to a source of osmotic stress with putative implications for plant–insect relations. Crop yield is a population trait, and in common to photosynthesis, plant–plant relations preclude scaling from plant to crop. Data-driven approaches require these theoretical insights, at the very least, to design more robust experiments and to make biological and agronomic sense of the wheat phenotype, the ultimate target of crop improvement.

Data accessibility. This article has no additional data.
Competing interests. I declare I have no competing interests.
Funding. I received no funding for this study.

## Endnote

[1]CIMMYT, the International Maize and Wheat Improvement Centre, was instrumental in the delivery of the semi-dwarf wheats at the core of the Green Revolution.

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
