## [Peer Review File · Proceedings of the Royal Society B: Biological Sciences]

Review History

RSPB-2021-1259.R0 (Original submission)

Review form: Reviewer 1

Recommendation

Reject – article is not of sufficient interest (we will consider a transfer to another journal)

Scientific importance: Is the manuscript an original and important contribution to its field?
Marginal

General interest: Is the paper of sufficient general interest?
Poor

Quality of the paper: Is the overall quality of the paper suitable?
Marginal

Is the length of the paper justified?
Yes

Should the paper be seen by a specialist statistical reviewer?
No

Do you have any concerns about statistical analyses in this paper? If so, please specify them explicitly in your report.

No

It is a condition of publication that authors make their supporting data, code and materials available - either as supplementary material or hosted in an external repository. Please rate, if applicable, the supporting data on the following criteria.

Is it accessible?

N/A

Is it clear?

N/A

Is it adequate?

N/A

Do you have any ethical concerns with this paper?

No

Comments to the Author

This paper purports to offer provide theoretical insights into the results of artificial selection from evolutionary and ecological biology, but these insights are not all firmly established. I shall concentrate on two: mother-offspring conflict, and trade-offs between root growth and grain number.

The evidence supposed to bear on mother-offspring conflict is in Fig. 2, which shows that wheat and salmon do not adjust offspring size when availability of resources varies. But this is not what mother-offspring conflict is about. Mother-offspring conflict is seen in evolutionary biology as occurring when genes are selected in offspring to obtain more resources from the mother than independent genes in the mother are selected to provide. The genes in the mother and in the offspring are considered as occurring at different loci, and expressed at different phenotypic ages. See Burt and Trivers: *Genes in Conflict*. It is not clear that the paper has anything to do with mother-offspring conflict.

In considering trade-offs between root growth and grain number, Section 3 of the paper shows that wheat plants allocate resources to reserves (WSC) as well as reproduction, so the allocation to reproduction is not fixed. The allocation to reserves increased in a five-decade selection experiment (Fig. 3d,e), and osmotic potential decreased as a correlated response (Fig. 3f). Section 4 shows that the high allocation to reserves may defend against aphids. While I see this may be interesting, it is does not seem unexpected to me.

Review form: Reviewer 2

Recommendation

Accept with minor revision (please list in comments)

Scientific importance: Is the manuscript an original and important contribution to its field?

Excellent

General interest: Is the paper of sufficient general interest?

Good

Quality of the paper: Is the overall quality of the paper suitable?

Excellent

Is the length of the paper justified?

Yes

Should the paper be seen by a specialist statistical reviewer?

No

Do you have any concerns about statistical analyses in this paper? If so, please specify them explicitly in your report.

No

It is a condition of publication that authors make their supporting data, code and materials available - either as supplementary material or hosted in an external repository. Please rate, if applicable, the supporting data on the following criteria.

Is it accessible?

N/A

Is it clear?

N/A

Is it adequate?

N/A

Do you have any ethical concerns with this paper?

No

Comments to the Author

A Darwinian view of the wheat phenotype; Victor Sadras; Proc Royal Soc

A refreshing view on phenotyping with some valuable new insights. Please see comments below

LINES 102-103

resources are difficult to estimate but crop growth rate in the critical period of grain set is a sound approximation [48, 50]

THIS IS A BIG ASSUMPTION AND IT SHOULD BE BETTER JUSTIFIED, DON'T EXPECT READERS NECESSARILY TO REFER BACK THE REVIEWS CITED.

108-114

EGR-GN is the efficiency of conversion of growth rate per unit grain size into grain number
Crops with more resources allocated to reserve will have lower efficiency to produce grain
Likewise, EGR-GN is expected to decline with increasing allocation of resources to root.

RESERVE AS WELL AS REMOBILIZATION NEED TO BE CONSIDERED. IT IS COMMON THAT IN A FAVORABLE ENVIRONMENT RESERVES REMAIN IN THE PLANT, AND SHOW GENOTYPE EFFECTS; IN A WATER- OR NUTRIENT-LIMITED SITUATION, RESERVE MOBILIZATION CERTAINLY BOOSTS YIELD, AS LATER STATED.

IS IT EGR-GN OR E (GR-GN)? IN ANY CASE I DID NOT SEE GR DEFINED

130-134

How, therefore, selection for yield and agronomic adaptation has favoured traits – amount and concentration of labile reserve carbohydrates – that are at best neutral for yield...correlates with

yet unrecognized, agronomically important traits.

THE LARGE GENETIC VARIATION EXPRESSED FOR RESERVES AMONG ELITE LINES -AT LEAST IN SPRING WHEAT- DOES SUGGEST TRADE-OFFS WITH OTHER PROCESSES, ROOTS BEING PERHAPS JUST ONE OF MANY.

156-159

This is illustrated in figure 4a, showing the relative growth rate of pea aphid (*Acyrtosiphon pisum*) was impaired by reduced feeding, reflecting the importance of sucrose as a phagostimulant at low dietary sucrose concentrations, and by osmoregulation failure at high concentrations

DID YOU MEAN TO SAY "IMPAIRED BY REDUCED FEEDING" ?

173-177

Empirical evidence linking labile reserve carbohydrates and aphid tolerance is just emerging, but the view of an ecological role of labile carbohydrates opens a new dimension to the established, narrow focus on plant carbon balance and source-sink relations.

AGREED, THIS GIVES NEW BREADTH TO THE TERM "ECO-PHYSIOLOGICAL TRAITS"

222-225

First, plant breeding is unlikely to improve traits shaped by natural selection over evolutionary time scales, such as the efficiency of photosynthetic enzymes [78, 87], but unrealised opportunities may exist for the selection of traits that increase crop yield at the expense of plant fitness [78, 88, 89] - plant breeding should be based on group selection [80].

NONETHELESS, EVEN PHOTOSYNTHETIC OR RESPIRATORY ENZYMES WILL RESPOND AS A FUNCTION OF THE MICRO-ENVIRONMENT OF A CANOPY -AFFECTING TEMPERATURE, LIGHT REGIME ETC., SO NATURAL VARIATION IN SUCH ENZYMES COULD BE SUBJECT TO SELECTION

225-226

Second, phenotyping and genotyping efforts must account for plant-plant relations [1, 2].

AGREED, ESPECIALLY AS THE MORE GENETICALLY COMPLEX THE TRAIT THE GREATER THE GxExM

Review form: Reviewer 3

Recommendation

Accept with minor revision (please list in comments)

Scientific importance: Is the manuscript an original and important contribution to its field?

Excellent

General interest: Is the paper of sufficient general interest?

Good

Quality of the paper: Is the overall quality of the paper suitable?

Excellent

Is the length of the paper justified?

Yes

Should the paper be seen by a specialist statistical reviewer?

No

Do you have any concerns about statistical analyses in this paper? If so, please specify them explicitly in your report.

No

It is a condition of publication that authors make their supporting data, code and materials available - either as supplementary material or hosted in an external repository. Please rate, if applicable, the supporting data on the following criteria.

Is it accessible?

N/A

Is it clear?

N/A

Is it adequate?

N/A

Do you have any ethical concerns with this paper?

No

Comments to the Author

This manuscript presents some innovative hypotheses, and the basic point: that agricultural research needs more theoretical developments if modern technologies and data are to be applied in useful ways, is certainly correct. I do see some minor issues, which I urge the author to consider.

The fact that plants produce a distribution of seed (here grain) sizes, and they vary the number of seeds produced in response to their conditions was documented by Harper in the 1970's and described in detail in his book on plant population biology - it is far from new. However, many agronomists will point out that this idea can be taken too far. Pruning experiments on beans (by G.E. Blackman if I remember correctly) over 50 years ago, showed that when all meristems were removed after flowering started so the plant could not grow more, plants did produce larger seeds when supplied with more resources. This is the "exception that proves the rule" that plants will do everything possible to respond with seed number rather than seed size. A more relevant example here is that, if a wheat field is fertilized early in the season, the plants will be larger and produce more grains, but if fertilization occurs after flowering when plants can no longer produce new flowering tillers, grains will be larger and have higher N content, which is important for yield quality.

I found the hypothesis that the high levels of WSC in in what plants is an evolved response to aphids intriguing, but not convincing. Have aphids been evolutionarily more important than other insect pests, especially chewing insects for which high levels of WSC would probably be a disadvantage? This hypothesis also suggests that much higher yield could be obtained if aphids are controlled and the level of WSC were used to increase yield. One would think that such behaviour would have appeared through breeding for high yield under high resource high pesticide applications - which represents much of the wheat breeding over the past 60 years. If there is a trade-off between WSC and root growth, it is not clear why there is no trade-off with yield. In any case, this is not a criticism of the manuscript. Agricultural research needs new ideas and hypotheses. Even if this hypothesis does not stand up to critical analyses, it can still be considered a useful contribution, because it will inspire alternative hypotheses.

Finally: What's in a title? For better or for worse, a title is important, and the current title is misleading. "Darwinian" is a now popular word in some agricultural research circles, and it has been promoted by Denison. But group selection, the basis of Denison's work and section 5 of the current manuscript, is actually anti-Darwinian. The same is true for earlier parts of the paper, in which population-level characteristics are considered drivers of yield production. For Darwin, evolution is driven by individuals and their offspring, although it is the population that evolves. Thus, while I have no objection to quoting Darwin throughout the paper, the title does not represent "truth in advertising". I would suggest something like "An evolutionary view..." or "An evolutionary-ecological view..." What is Darwinian here is the change in emphasis from the plant and its abiotic environment (which Harper called "Wallacian") to the plant and its biotic environment (which he called Darwinian). This change in emphasis is long overdue, especially in crop physiology.

Some minor comments:

I. 46 "abiotic factors" would be a better term than "the elements" and "respond to" would be less adaptationist than "accommodate" (also in Fig. 1 legend)

I. 54 remove "this is universal" - it is not. Some horticultural crops have been selected to not overproduce flowers. I'm sure there are also exceptions in wild plants. Such universality is not common in the plant kingdom.

I. 59 "...we observe that annual..."

I. 61 replace ";" with "and".

I. 92-94 The emphasis on resources is totally appropriate, but conditions, as well as resources, can limit growth.

II. 95-98 - "The first assumption is that plants are able to account for past, current and future environmental conditions by combining memory mechanisms including epigenetics, proximate environmental cues such as direct sensing of water and nutrient availability in soil, and cues such as photoperiod that allow for future conditions..." This is extremely adaptationist and expects more of crop plants than is reasonable. Plants are not omnipotent, as this sentence implies. Constraints are as important as abilities in agriculture.

Decision letter (RSPB-2021-1259.R0)

06-Jul-2021

Dear Dr Sadras:

Your manuscript has now been peer reviewed and the reviewers' comments (not including confidential comments to the Editor) are included at the end of this email for your reference. As you will see, the reviewers have raised some concerns with your manuscript - for one of these (referee 1) the points are sufficiently serious to merit a recommendation of rejection but referees 2 and 3 consider that their points could be addressed in a revision. My view is that the necessary changes are potentially achievable without this being a totally new manuscript, so I would like to give you the chance to revise your manuscript. And I do agree with referee 3 about the title.

We do not allow multiple rounds of revision so we urge you to make every effort to fully address all of the comments at this stage. If deemed necessary by the Associate Editor, your manuscript will be sent back to one or more of the original reviewers for assessment. If the original reviewers

are not available we may invite new reviewers. Please note that we cannot guarantee eventual acceptance of your manuscript at this stage.

Research ethics:

Use of animals and field studies:

It is a condition of publication that you make available the data and research materials supporting the results in the article (<https://royalsociety.org/journals/authors/author-guidelines/#data>). Datasets should be deposited in an appropriate publicly available repository and details of the associated accession number, link or DOI to the datasets must be included in the Data Accessibility section of the article (<https://royalsociety.org/journals/ethics-policies/data-sharing-mining/>). Reference(s) to datasets should also be included in the reference list of the article with DOIs (where available).

Please submit a copy of your revised paper within three weeks. If we do not hear from you within this time your manuscript will be rejected. If you are unable to meet this deadline please let us know as soon as possible, as we may be able to grant a short extension.

Best wishes,
Innes Cuthill

Prof. Innes Cuthill
Reviews Editor
mailto:proceedingsb@royalsociety.org

Reviewer(s)' Comments to Author:

Referee: 1

Comments to the Author(s)

This paper purports to offer provide theoretical insights into the results of artificial selection from evolutionary and ecological biology, but these insights are not all firmly established. I shall concentrate on two: mother-offspring conflict, and trade-offs between root growth and grain number.

The evidence supposed to bear on mother-offspring conflict is in Fig. 2, which shows that wheat and salmon do not adjust offspring size when availability of resources varies. But this is not what mother-offspring conflict is about. Mother-offspring conflict is seen in evolutionary biology as occurring when genes are selected in offspring to obtain more resources from the mother than independent genes in the mother are selected to provide. The genes in the mother and in the offspring are considered as occurring at different loci, and expressed at different phenotypic ages. See Burt and Trivers: Genes in Conflict. It is not clear that the paper has anything to do with mother-offspring conflict.

In considering trade-offs between root growth and grain number, Section 3 of the paper shows that wheat plants allocate resources to reserves (WSC) as well as reproduction, so the allocation to reproduction is not fixed. The allocation to reserves increased in a five-decade selection experiment (Fig. 3d,e), and osmotic potential decreased as a correlated response (Fig. 3f). Section 4 shows that the high allocation to reserves may defend against aphids. While I see this may be interesting, it does not seem unexpected to me.

Referee: 2

Comments to the Author(s)

A Darwinian view of the wheat phenotype; Victor Sadras; Proc Royal Soc

A refreshing view on phenotyping with some valuable new insights. Please see comments below

LINES 102-103

resources are difficult to estimate but crop growth rate in the critical period of grain set is a sound approximation [48, 50]

THIS IS A BIG ASSUMPTION AND IT SHOULD BE BETTER JUSTIFIED, DON'T EXPECT READERS NECESSARILY TO REFER BACK THE REVIEWS CITED.

108-114

EGR-GN is the efficiency of conversion of growth rate per unit grain size into grain number
Crops with more resources allocated to reserve will have lower efficiency to produce grain
Likewise, EGR-GN is expected to decline with increasing allocation of resources to root.

RESERVE AS WELL AS REMOBILIZATION NEED TO BE CONSIDERED. IT IS COMMON THAT IN A FAVORABLE ENVIRONMENT RESERVES REMAIN IN THE PLANT, AND SHOW GENOTYPE EFFECTS; IN A WATER- OR NUTRIENT-LIMITED SITUATION, RESERVE MOBILIZATION CERTAINLY BOOSTS YIELD, AS LATER STATED.

IS IT EGR-GN OR E (GR-GN)? IN ANY CASE I DID NOT SEE GR DEFINED

130-134

How, therefore, selection for yield and agronomic adaptation has favoured traits – amount and concentration of labile reserve carbohydrates – that are at best neutral for yield...correlates with yet unrecognized, agronomically important traits.

THE LARGE GENETIC VARIATION EXPRESSED FOR RESERVES AMONG ELITE LINES -AT LEAST IN SPRING WHEAT- DOES SUGGEST TRADE-OFFS WITH OTHER PROCESSES, ROOTS BEING PERHAPS JUST ONE OF MANY.

156-159

This is illustrated in figure 4a, showing the relative growth rate of pea aphid (*Acyrtosiphon pisum*) was impaired by reduced feeding, reflecting the importance of sucrose as a phagostimulant at low dietary sucrose concentrations, and by osmoregulation failure at high concentrations

DID YOU MEAN TO SAY "IMPAIRED BY REDUCED FEEDING" ?

173-177

Empirical evidence linking labile reserve carbohydrates and aphid tolerance is just emerging, but the view of an ecological role of labile carbohydrates opens a new dimension to the established, narrow focus on plant carbon balance and source-sink relations.

AGREED, THIS GIVES NEW BREADTH TO THE TERM "ECO-PHYSIOLOGICAL TRAITS"

222-225

First, plant breeding is unlikely to improve traits shaped by natural selection over evolutionary time scales, such as the efficiency of photosynthetic enzymes [78, 87], but unrealised opportunities may exist for the selection of traits that increase crop yield at the expense of plant fitness [78, 88, 89] – plant breeding should be based on group selection [80].

NONETHELESS, EVEN PHOTOSYNTHETIC OR RESPIRATORY ENZYMES WILL RESPOND AS A FUNCTION OF THE MICRO-ENVIRONMENT OF A CANOPY -AFFECTING TEMPERATURE, LIGHT REGIME ETC., SO NATURAL VARIATION IN SUCH ENZYMES COULD BE SUBJECT TO SELECTION

225-226

Second, phenotyping and genotyping efforts must account for plant-plant relations [1, 2].
AGREED, ESPECIALLY AS THE MORE GENETICALLY COMPLEX THE TRAIT THE GREATER THE GxExM

Referee: 3

Comments to the Author(s)

This manuscript presents some innovative hypotheses, and the basic point: that agricultural research needs more theoretical developments if modern technologies and data are to be applied in useful ways, is certainly correct. I do see some minor issues, which I urge the author to consider.

The fact that plants produce a distribution of seed (here grain) sizes, and they vary the number of seeds produced in response to their conditions was documented by Harper in the 1970's and described in detail in his book on plant population biology - it is far from new. However, many agronomists will point out that this idea can be taken too far. Pruning experiments on beans (by G.E. Blackman if I remember correctly) over 50 years ago, showed that when all meristems were removed after flowering started so the plant could not grow more, plants did produce larger seeds when supplied with more resources. This is the "exception that proves the rule" that plants will do everything possible to respond with seed number rather than seed size. A more relevant example here is that, if a wheat field is fertilized early in the season, the plants will be larger and produce more grains, but if fertilization occurs after flowering when plants can no longer produce new flowering tillers, grains will be larger and have higher N content, which is important for yield quality.

I found the hypothesis that the high levels of WSC in in what plants is an evolved response to aphids intriguing, but not convincing. Have aphids been evolutionarily more important than other insect pests, especially chewing insects for which high levels of WSC would probably be a disadvantage? This hypothesis also suggests that much higher yield could be obtained if aphids are controlled and the level of WSC were used to increase yield. One would think that such behaviour would have appeared through breeding for high yield under high resource high pesticide applications - which represents much of the wheat breeding over the past 60 years. If there is a trade-off between WSC and root growth, it is not clear why there is no trade-off with yield. In any case, this is not a criticism of the manuscript. Agricultural research needs new ideas and hypotheses. Even if this hypothesis does not stand up to critical analyses, it can still be considered a useful contribution, because it will inspire alternative hypotheses.

Finally: What's in a title? For better or for worse, a title is important, and the current title is misleading. "Darwinian" is a now popular word in some agricultural research circles, and it has been promoted by Denison. But group selection, the basis of Denison's work and section 5 of the current manuscript, is actually anti-Darwinian. The same is true for earlier parts of the paper, in which population-level characteristics are considered drivers of yield production. For Darwin, evolution is driven by individuals and their offspring, although it is the population that evolves. Thus, while I have no objection to quoting Darwin throughout the paper, the title does not represent "truth in advertising". I would suggest something like "An evolutionary view..." or "An evolutionary-ecological view..." What is Darwinian here is the change in emphasis from the plant and its abiotic environment (which Harper called "Wallacian") to the plant and its biotic environment (which he called Darwinian). This change in emphasis is long overdue, especially in crop physiology.

Some minor comments:

l. 46 "abiotic factors" would be a better term than "the elements" and "respond to" would be less adaptationist than "accommodate" (also in Fig. 1 legend)

l. 54 remove "this is universal" - it is not. Some horticultural crops have been selected to not overproduce flowers. I'm sure there are also exceptions in wild plants. Such universality is not common in the plant kingdom.

l. 59 "...we observe that annual..."

I. 61 replace ";" with "and".

I. 92-94 The emphasis on resources is totally appropriate, but conditions, as well as resources, can limit growth.

II. 95-98 - "The first assumption is that plants are able to account for past, current and future environmental conditions by combining memory mechanisms including epigenetics, proximate environmental cues such as direct sensing of water and nutrient availability in soil, and cues such as photoperiod that allow for future conditions..." This is extremely adaptationist and expects more of crop plants than is reasonable. Plants are not omnipotent, as this sentence implies. Constraints are as important as abilities in agriculture.

Author's Response to Decision Letter for (RSPB-2021-1259.R0)

See Appendix A.

RSPB-2021-1259.R1 (Revision)

Review form: Reviewer 2

Recommendation

Accept with minor revision (please list in comments)

Scientific importance: Is the manuscript an original and important contribution to its field?

Excellent

General interest: Is the paper of sufficient general interest?

Good

Quality of the paper: Is the overall quality of the paper suitable?

Excellent

Is the length of the paper justified?

Yes

Should the paper be seen by a specialist statistical reviewer?

No

Do you have any concerns about statistical analyses in this paper? If so, please specify them explicitly in your report.

No

It is a condition of publication that authors make their supporting data, code and materials available - either as supplementary material or hosted in an external repository. Please rate, if applicable, the supporting data on the following criteria.

Is it accessible?

N/A

Is it clear?

N/A

Is it adequate?

N/A

Do you have any ethical concerns with this paper?

No

Comments to the Author

All good now, except I suggest the response to the comment below be to make it explicit that even enzymes shaped by natural selection may need to be modified -or genetic variation tapped- to fits the unique crop canopy environment. That was at best ambiguous in the original statement.

222-225 First, plant breeding is unlikely to improve traits shaped by natural selection over evolutionary time scales, such as the efficiency of photosynthetic enzymes [78, 87], but unrealised opportunities may exist for the selection of traits that increase crop yield at the expense of plant fitness [78, 88, 89] - plant breeding should be based on group selection [80].

NONETHELESS, EVEN PHOTOSYNTHETIC OR RESPIRATORY ENZYMES WILL RESPOND AS A FUNCTION OF THE MICRO-ENVIRONMENT OF A CANOPY -AFFECTING TEMPERATURE, LIGHT REGIME ETC., SO NATURAL VARIATION IN SUCH ENZYMES COULD BE SUBJECT TO SELECTION. Agree and this is captured in the original statement with supporting references; no change.

Decision letter (RSPB-2021-1259.R1)

11-Aug-2021

Dear Dr Sadras

I am pleased to inform you that your manuscript RSPB-2021-1259.R1 entitled "Evolutionary and ecological perspectives on the wheat phenotype" has been accepted for publication in Proceedings B.

The referee is happy with your revisions and has recommended publication, but also suggests a further minor revision to your manuscript. Therefore, I invite you to respond to the comments and revise your manuscript. Because the schedule for publication is very tight, it is a condition of publication that you submit the revised version of your manuscript within 7 days. If you do not think you will be able to meet this date please let us know.

When submitting your revised manuscript, you will be able to respond to the comments made by the referee(s) and upload a file "Response to Referees". You can use this to document any changes you make to the original manuscript. We require a copy of the manuscript with revisions made

since the previous version marked as 'tracked changes' to be included in the 'response to referees' document.

Best wishes,
Innes Cuthill

Prof. Innes Cuthill
Reviews Editor, Proceedings B
mailto: proceedingsb@royalsociety.org

Reviewer(s)' Comments to Author:

Referee: 2

Comments to the Author(s)

All good now, except I suggest the response to the comment below be to make it explicit that even enzymes shaped by natural selection may need to be modified -or genetic variation tapped- to fits the unique crop canopy environment. That was at best ambiguous in the original statement.

222-225 First, plant breeding is unlikely to improve traits shaped by natural selection over evolutionary time scales, such as the efficiency of photosynthetic enzymes [78, 87], but unrealised opportunities may exist for the selection of traits that increase crop yield at the expense of plant fitness [78, 88, 89] - plant breeding should be based on group selection [80].

NONETHELESS, EVEN PHOTOSYNTHETIC OR RESPIRATORY ENZYMES WILL RESPOND AS A FUNCTION OF THE MICRO-ENVIRONMENT OF A CANOPY -AFFECTING TEMPERATURE, LIGHT REGIME ETC., SO NATURAL VARIATION IN SUCH ENZYMES COULD BE SUBJECT TO SELECTION. Agree and this is captured in the original statement with supporting references; no change.

Author's Response to Decision Letter for (RSPB-2021-1259.R1)

See Appendix B.

Decision letter (RSPB-2021-1259.R2)

12-Aug-2021

Dear Dr Sadras

I am pleased to inform you that your manuscript entitled "Evolutionary and ecological perspectives on the wheat phenotype" has been accepted for publication in Proceedings B.

If you are likely to be away from e-mail contact during this period, let us know. Due to rapid publication and an extremely tight schedule, if comments are not received, we may publish the paper as it stands.

Data Accessibility section

Open access

You are invited to opt for open access via our author pays publishing model. Payment of open access fees will enable your article to be made freely available via the Royal Society website as soon as it is ready for publication. For more information about open access publishing please visit our website at http://royalsocietypublishing.org/site/authors/open_access.xhtml.

The open access fee is £1,700 per article (plus VAT for authors within the EU). If you wish to opt for open access then please let us know as soon as possible.

Paper charges

Sincerely,
Proceedings B
<mailto:proceedingsb@royalsociety.org>

Appendix A

Prof. Innes Cuthill
Reviews Editor

Thanks for your editorial input, reports of three reviewers, and the opportunity to submit a revised version of my article.

I have addressed all comments as explained in the question-answer section below. The title has been changed as proposed. Key words were updated to account for the change in title. The most important change relates to Reviewer 1 comment on mother-offspring conflict. The original ms relied excessively on references. To correct this, the revised ms (i) spells out the mother-offspring conflict and (ii) includes a new Fig. 2b. The revised version is 3100 words including abstract.

Below please also find the revised ms with track changes.

Sincerely,

Victor Sadras

Referee: 1

Comments to the Author(s)

This paper purports to offer provide theoretical insights into the results of artificial selection from evolutionary and ecological biology, but these insights are not all firmly established. I shall concentrate on two: mother-offspring conflict, and trade-offs between root growth and grain number.

The evidence supposed to bear on mother-offspring conflict is in Fig. 2, which shows that wheat and salmon do not adjust offspring size when availability of resources varies. But this is not what mother-offspring conflict is about. Mother-offspring conflict is seen in evolutionary biology as occurring when genes are selected in offspring to obtain more resources from the mother than independent genes in the mother are selected to provide. The genes in the mother and in the offspring are considered as occurring at different loci, and expressed at different phenotypic ages. See Burt and Trivers: Genes in Conflict. It is not clear that the paper has anything to do with mother-offspring conflict. **Agree with this comment; Fig. 2 and associated text do not show mother-offspring conflict. The original ms relied excessively on references. To correct this, the revised ms (i) spells out the mother-offspring conflict with supporting references, and (ii) includes new Fig. 2b.**

In considering trade-offs between root growth and grain number, Section 3 of the paper shows that wheat plants allocate resources to reserves (WSC) as well as reproduction, so the allocation to reproduction is not fixed. The allocation to reserves increased in a five-decade selection experiment (Fig. 3d,e), and osmotic potential decreased as a correlated response (Fig. 3f). Section 4 shows that the high allocation to reserves may defend against aphids. While I see this may be interesting, it is does not seem unexpected to me. **Thanks for comment and good summary. Here the point is that the dominant view in plant breeding has been that reserves buffer grain size, and therefore efforts have been allocated to incorporate WSC in breeding programs – crosses were done, populations screened, and QTL identified. Del Pozo in Chile and Ovenden in Australia are examples of massive investments of money and time along these lines (del Pozo et al., 2016; Ovenden et al., 2017). The rationale in Ovenden et al reads: “The potential to increase the genetic capacity for water-soluble carbohydrate (WSC) accumulation is an opportunity to improve the drought tolerance capability of rainfed wheat varieties...” These large investments have failed because the model of the phenotype was incomplete: trade-offs (with grain number and root biomass) and osmotic effects were completely overlooked. This is a primary motivation of this article as stated in introduction and conclusion.**

Referee: 2

Comments to the Author(s)

A Darwinian view of the wheat phenotype; Victor Sadras; Proc Royal Soc

A refreshing view on phenotyping with some valuable new insights. **Thanks.**

Please see comments below

LINES 102-103

resources are difficult to estimate but crop growth rate in the critical period of grain set is a sound approximation [48, 50] THIS IS A BIG ASSUMPTION AND IT SHOULD BE BETTER JUSTIFIED, DON'T EXPECT READERS NECESSARILY TO REFER BACK THE REVIEWS CITED. **The revised ms expands this sentence and provides further supporting references. This assumption is the basis of mathematical models of crop yield.**

The relationship between seed number and growth rate has been verified in wheat, pulses, soybean, maize, sunflower. The graph below is typical. Please also see Fig. 3a supporting the strong correlation growth rate-grain number.

108-114

EGR-GN is the efficiency of conversion of growth rate per unit grain size into grain number
Crops with more resources allocated to reserve will have lower efficiency to produce grain
Likewise, EGR-GN is expected to decline with increasing allocation of resources to root.

RESERVE AS WELL AS REMOBILIZATION NEED TO BE CONSIDERED. IT IS COMMON THAT IN A FAVORABLE ENVIRONMENT RESERVES REMAIN IN THE PLANT, AND SHOW GENOTYPE EFFECTS; IN A WATER- OR NUTRIENT-LIMITED SITUATION, RESERVE MOBILIZATION CERTAINLY BOOSTS YIELD, AS LATER STATED. This is a good point; in good growing conditions a large amount of WSC can remain in the crop. The revised ms includes a statement and new supporting reference in this regard.

IS IT EGR-GN OR E (GR-GN)? IN ANY CASE I DID NOT SEE GR DEFINED. It is E (for efficiency) with subscript GR-GN indicating the conversion from growth rate to grain number. Corrected in revised ms.

130-134

How, therefore, selection for yield and agronomic adaptation has favoured traits – amount and concentration of labile reserve carbohydrates – that are at best neutral for yield...correlates with yet unrecognized, agronomically important traits.

THE LARGE GENETIC VARIATION EXPRESSED FOR RESERVES AMONG ELITE LINES -AT LEAST IN SPRING WHEAT- DOES SUGGEST TRADE-OFFS WITH OTHER PROCESSES, ROOTS BEING PERHAPS JUST ONE OF MANY. OK; the ms focuses on the two trade-offs that are (i) supported by empirical evidence, and (ii) functionally relevant. I rather don't speculate on other trade-offs.

156-159

This is illustrated in figure 4a, showing the relative growth rate of pea aphid (*Acyrtosiphon pisum*) was impaired by reduced feeding, reflecting the importance of sucrose as a phagostimulant at low dietary sucrose concentrations, and by osmoregulation failure at high concentrations
DID YOU MEAN TO SAY "IMPAIRED BY REDUCED FEEDING" ? **Original ms was correct, no changes.**

173-177

Empirical evidence linking labile reserve carbohydrates and aphid tolerance is just emerging, but the view of an ecological role of labile carbohydrates opens a new dimension to the established, narrow focus on plant carbon balance and source-sink relations.

AGREED, THIS GIVES NEW BREADTH TO THE TERM "ECO-PHYSIOLOGICAL TRAITS". **OK, no change in response to this comment.**

222-225

First, plant breeding is unlikely to improve traits shaped by natural selection over evolutionary time scales, such as the efficiency of photosynthetic enzymes [78, 87], but unrealised opportunities may exist for the selection of traits that increase crop yield at the expense of plant fitness [78, 88, 89] – plant breeding should be based on group selection [80].

NONETHELESS, EVEN PHOTOSYNTHETIC OR RESPIRATORY ENZYMES WILL RESPOND AS A FUNCTION OF THE MICRO-ENVIRONMENT OF A CANOPY -AFFECTING TEMPERATURE, LIGHT REGIME ETC., SO NATURAL VARIATION IN SUCH ENZYMES COULD BE SUBJECT TO SELECTION. **Agree and this is captured in the original statement with supporting references; no change.**

225-226

Second, phenotyping and genotyping efforts must account for plant-plant relations [1, 2].

AGREED, ESPECIALLY AS THE MORE GENETICALLY COMPLEX THE TRAIT THE GREATER THE GxExM. **OK**

Referee: 3

Comments to the Author(s)

This manuscript presents some innovative hypotheses, and the basic point: that agricultural research needs more theoretical developments if modern technologies and data are to be applied in useful ways, is certainly correct. I do see some minor issues, which I urge the author to consider.

The fact that plants produce a distribution of seed (here grain) sizes, and they vary the number of seeds produced in response to their conditions was documented by Harper in the 1970's and described in detail in his book on plant population biology - it is far from new. However, many agronomists will point out that this idea can be taken too far. Pruning experiments on beans (by G.E. Blackman if I remember correctly) over 50 years ago, showed that when all meristems were removed after flowering started so the plant could not grow more, plants did produce larger seeds when supplied with more resources. This is the "exception that proves the rule" that plants will do everything possible to respond with seed number rather than seed size. A more relevant example here is that, if a wheat field is fertilized early in the season, the plants will be larger and produce more grains, but if fertilization occurs after flowering when plants can no longer produce new flowering tillers, grains will be larger and have higher N content, which is important for yield quality. **OK. I completely agree the number-size trade-off is not new. The point of discussion is the interpretation of the correlation yield-grain number, and the shift from misconstrued cause-and-effect to a broader view informed by evolutionary perspective. This was expanded in the revised ms.**

I found the hypothesis that the high levels of WSC in in what plants is an evolved response to aphids intriguing, but not convincing. Have aphids been evolutionarily more important than other insect pests, especially chewing insects for which high levels of WSC would probably be a disadvantage? This hypothesis also suggests that much higher yield could be obtained if aphids are controlled and the level of WSC were used to increase yield. One would think that such behaviour would have appeared through breeding for high yield under high resource high pesticide applications - which represents much of the wheat breeding over the past 60 years. If there is a trade-off between WSC and root growth, it is not clear why there is no trade-off with yield. In any case, this is not a criticism of the manuscript. Agricultural research needs new ideas and hypotheses. Even if this hypothesis does not stand up to critical analyses, it can still be considered a useful contribution, because it will inspire alternative hypotheses. **The reviewer captured well the spirit of this section. I accept the link aphids-WSC is far from proved, and this is explicit in the paper where it reads: "Empirical evidence linking labile reserve carbohydrates and aphid tolerance is just emerging, but the view of an ecological role of labile carbohydrates opens a new dimension to the established, narrow focus on plant carbon balance and source-sink relations."** The point is the need for new ideas. We cannot discard the role of other herbivores as pointed out by the reviewer, but the case for aphids is supported by: (a) aphids and the viruses they carry are the major pests of wheat and a focus of breeding programs worldwide and (b) intricate mechanisms for osmoregulation in aphids.

Finally: What's in a title? For better or for worse, a title is important, and the current title is misleading. "Darwinian" is a now popular word in some agricultural research circles, and it has been promoted by Denison. But group selection, the basis of Denison's work and section 5 of the current manuscript, is actually anti-Darwinian. The same is true for earlier parts of the paper, in which population-level characteristics are considered drivers of yield production. For Darwin, evolution is driven by individuals and their offspring, although it is the population that evolves. Thus, while I have no objection to quoting Darwin throughout the paper, the title does not represent "truth in advertising". I would suggest something like "An evolutionary view..." or "An evolutionary-ecological view..." What is Darwinian here is the change in emphasis from the plant and its abiotic environment (which Harper called "Wallacian") to the plant and its biotic environment (which he called Darwinian). This change in emphasis is long overdue, especially in crop physiology.

ACCEPTED. The title was changed as suggested to "Evolutionary and ecological perspectives on the wheat phenotype"

Some minor comments:

I. 46 "abiotic factors" would be a better term than "the elements" and "respond to" would be less adaptationist than "accommodate" (also in Fig. 1 legend). **OK changed as proposed.**

I. 54 remove "this is universal" - it is not. Some horticultural crops have been selected to not overproduce flowers. I'm sure there are also exceptions in wild plants. Such universality is not common in the plant kingdom. **OK, deleted.**

I. 59 "...we observe that annual..." **OK, changed as indicated.**

I. 61 replace ";" with "and". **I think "and" reads better here, no change.**

I. 92-94 The emphasis on resources is totally appropriate, but conditions, as well as resources, can limit growth. **Agree. Here we assume that the plant has R resources (e.g. total C), which result from**

environmental resources and conditions. Not unlike Smith and Fretwell, we ask how are R resources allocated. No changes.

II. 95-98 - "The first assumption is that plants are able to account for past, current and future environmental conditions by combining memory mechanisms including epigenetics, proximate environmental cues such as direct sensing of water and nutrient availability in soil, and cues such as photoperiod that allow for future conditions..." This is extremely adaptationist and expects more of crop plants than is reasonable. Plants are not omnipotent, as this sentence implies. Constraints are as important as abilities in agriculture. **OK, I re-wrote this sentence to avoid the interpretation of omnipotent plant; it was not the intention.**

del Pozo, A. et al., 2016. Physiological Traits Associated with Wheat Yield Potential and Performance under Water-Stress in a Mediterranean Environment. *Frontiers in Plant Science*, 7.

Ovenden, B. et al., 2017. Selection for water-soluble carbohydrate accumulation and investigation of genetic x environment interactions in an elite wheat breeding population. *Theor. Appl. Gen.*, 130(11): 2445-2461.

Smith, C.C. and Fretwell, S.D., 1974. The optimal balance between size and number of offspring. *Am. Nat.*, 108: 499-506.

Appendix B

Comments to the Author(s)

All good now, except I suggest the response to the comment below be to make it explicit that even enzymes shaped by natural selection may need to be modified -or genetic variation tapped- to fit the unique crop canopy environment. That was at best ambiguous in the original statement.

222-225 First, plant breeding is unlikely to improve traits shaped by natural selection over evolutionary time scales, such as the efficiency of photosynthetic enzymes [78, 87], but unrealised opportunities may exist for the selection of traits that increase crop yield at the expense of plant fitness [78, 88, 89] – plant breeding should be based on group selection [80].

NONETHELESS, EVEN PHOTOSYNTHETIC OR RESPIRATORY ENZYMES WILL RESPOND AS A FUNCTION OF THE MICRO-ENVIRONMENT OF A CANOPY -AFFECTING TEMPERATURE, LIGHT REGIME ETC., SO NATURAL VARIATION IN SUCH ENZYMES COULD BE SUBJECT TO SELECTION. Agree and this is captured in the original statement with supporting references; no change.

In response to this observation, the revised ms expanded the paragraph in lines 216-225 to make explicit that even enzymes shaped by natural selection may need to be modified -or genetic variation tapped- to fit the current crop environment.